# Electrically tunable organic–inorganic hybrid polaritons with monolayer WS$_2$

Lucas C. Flatten[1], David M. Coles[1,2], Zhengyu He[1], David G. Lidzey[3], Robert A. Taylor[2], Jamie H. Warner[1] & Jason M. Smith[1]

Exciton-polaritons are quasiparticles consisting of a linear superposition of photonic and excitonic states, offering potential for nonlinear optical devices. The excitonic component of the polariton provides a finite Coulomb scattering cross section, such that the different types of exciton found in organic materials (Frenkel) and inorganic materials (Wannier-Mott) produce polaritons with different interparticle interaction strength. A hybrid polariton state with distinct excitons provides a potential technological route towards *in situ* control of nonlinear behaviour. Here we demonstrate a device in which hybrid polaritons are displayed at ambient temperatures, the excitonic component of which is part Frenkel and part Wannier-Mott, and in which the dominant exciton type can be switched with an applied voltage. The device consists of an open microcavity containing both organic dye and a monolayer of the transition metal dichalcogenide WS$_2$. Our findings offer a perspective for electrically controlled nonlinear polariton devices at room temperature.

[1] Department of Materials, University of Oxford, Parks Road, Oxford OX1 3PH, UK. [2] Department of Physics, Clarendon Laboratory, University of Oxford, Oxford OX1 3PU, UK. [3] Department of Physics & Astronomy, University of Sheffield, Sheffield S3 7RH, UK. Correspondence and requests for materials should be addressed to L.C.F. (email: lucas.flatten@materials.ox.ac.uk) or to J.M.S. (email: jason.smith@materials.ox.ac.uk).

Exciton-polaritons result from strongly coupling an optical cavity mode to electronic transitions with reversible energy exchange between the two[1,2]. Polaritons inherit properties such as the delocalized photonic wavefunction and finite excitonic mass from their constituents. Nonlinearities in an ensemble of polaritons arise as a result of the Coulomb interaction via the excitonic fraction of the state[3,4]. Since the first demonstration of exciton-polaritons[1], strongly correlated effects such as Bose-Einstein condensation in a semiconductor quantum-well microcavity at cryogenic temperatures[5] and more recently with a polymer at room temperature[6] have been shown. Superfluidity[7], polariton lasing[8] and multistability[9] have furthered the range of nonlinear phenomena these ultra-light bosonic quasiparticles display. The exciton part of polaritons plays a vital role as it conveys the interparticle interaction potential. Generally two different types of excitons are distinguished. The Frenkel exciton is characterized by its strong binding energy (order of $1\,eV$) and large oscillator strength[10]. However, a small Bohr radius ($\sim 1\,nm$) and relatively low mobility ($\sim 10^{-2}\,cm^2\,V^{-1}\,s^{-1}$) make for weak exciton–exciton interaction cross sections. Wannier-Mott (WM) excitons generally have lower binding energies (a few meV) and smaller oscillator strengths but larger scattering cross sections. For the recently emerged class of two-dimensional transition metal dichalcogenides (TMD) the exciton classification is not trivial since the exciton binding energy can be as high as $700\,meV$ (ref. 11). However, due to their crystalline structure the carrier mobility is large ($\sim 40\,cm^2\,V^{-1}\,s^{-1}$)[12], the Coulomb interaction in the confined plane is strong and the wavefunction is of WM type (that is, with an extension over a large number of unit cells)[13]. TMD monolayers are direct bandgap materials that display intriguing optical properties potentially useful for future applications and fundamental research, such as high quantum yields and strong absorption in the visible, enhanced excitonic Coulomb interactions and valley degrees of freedom[12–15]. Through a proper choice of substrate and the reduction of impurities the mobilities in such monolayers can be enhanced markedly and electrical charge injection is possible[15,16].

Engineering polariton states for nonlinear effects in compact optoelectronic devices remains a major challenge. A possible pathway towards controlling the physical properties of polaritons within a device is the use of hybrid polariton states[17,18]. Such states combine excitonic properties from different materials, thus creating polaritons with properties beyond those that can be created by any single material. The superposition weight of their components can be modified in situ by changing the resonance condition between the different excitons and the cavity mode.

In this article we demonstrate the formation of hybrid organic-inorganic polaritons created through the simultaneous coupling of the J-aggregate dye TDBC and a tungsten-disulphide ($WS_2$) monolayer to a confined optical microcavity mode[19,20]. The cavity consists of a distributed Bragg reflector (DBR) with $WS_2$ flakes on the low refractive index terminated side and a small silver mirror covered with a thin layer of the organic dye (Fig. 1a–c). When the cavity mode energy is tuned between the two exciton energies, it couples simultaneously to both exciton types thereby creating hybrid organic–inorganic polariton states. The $WS_2$ layer is placed between two electrodes such that an applied voltage perturbs the energy of the WM excitons and thereby modifies the composition of the hybrid polariton state. The change in the spectral position alters the relative mixing of Frenkel and WM excitons within polariton states, thus allowing for in situ control of polariton properties such as mobility and scattering cross section.

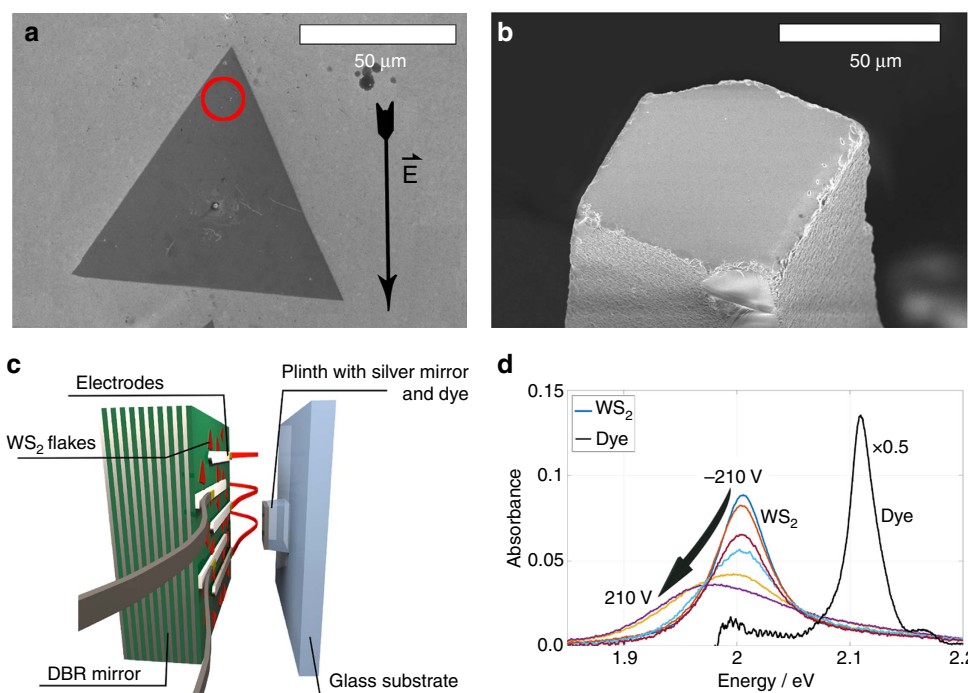

**Figure 1 | Two-dimensional $WS_2$ between electrodes in an optical microcavity.** (**a**) SEM-micrograph of a monolayer $WS_2$ flake deposited on $SiO_2$ terminated DBR, forming one side of the optical microcavity. The arrow denotes the direction of the applied electric field and the red circle the area from which the data were obtained. (**b**) SEM image of the opposing cavity side, a silver mirror on a silica plinth. (**c**) Sketch of the two mirrors with silver electrodes on the surface of the DBR giving electrical tunability within the cavity. (**d**) Absorbance of atomically-thin $WS_2$ for various applied voltages obtained from the position marked with a red circle in a (colour) and the organic dye TDBC (black, scaled by factor 0.5).

## Results

**Hybrid polariton states.** Both the organic dye and the inorganic WS$_2$ monolayer absorb strongly in the visible, with narrow, excitonic absorption peaks at $E_F = 2.11$ eV and $E_{WM} = 2.01$ eV, respectively. Placed in an electric field, the WS$_2$ absorption peak position is additionally tunable within $\sim 20$ meV to the red for electric field strengths of $E = 2.33 \times 10^4$ V cm$^{-1}$ (Fig. 1d). This effect is caused by changes in the local electron density which causes a shift in spectral weight of neutral ($X^0$) and charged excitons ($X^-$) and has been discussed elsewhere[14,21–23]. Here we use the effect to change the WS$_2$ absorption peak position, amplitude and width within the cavity to alter a polariton state. This polariton state is of hybrid nature, combining two different species of excitons, WM excitons formed in the inorganic component WS$_2$ and Frenkel (F) excitons in the organic J-aggregated dye TDBC. The system of one cavity mode simultaneously coupled to two excitonic transitions is described by the Hamiltonian

$$H = E_c b^\dagger b + E_F x_F^\dagger x_F + E_{WM} x_{WM}^\dagger x_{WM} + \\ V_F \left( b^\dagger x_F + b x_F^\dagger \right) + V_{WM} \left( b^\dagger x_{WM} + b x_{WM}^\dagger \right) \quad (1)$$

where $V_F$ and $V_{WM}$ are the interaction potentials between the cavity mode and F- and WM excitons. $b$, $x_F$ and $x_{WM}$ are the photon, F- and WM-exciton annihilation operators respectively (here for one $k$ vector only). In the stationary case the system can

be reduced to:

$$H|\Psi\rangle = \begin{pmatrix} E_c & V_F & V_{WM} \\ V_F & E_F & 0 \\ V_{WM} & 0 & E_{WM} \end{pmatrix} \begin{pmatrix} \alpha \\ \beta \\ \gamma \end{pmatrix} = E|\Psi\rangle \quad (2)$$

Here the state $|\Psi\rangle$ is defined by the three coefficients $\alpha$, $\beta$ and $\gamma$, which quantify the contribution of photon, F- and WM-exciton respectively. Each of the three eigenstates forms a polariton branch, whose photonic part can be readily observed spectroscopically. Figure 2a–c show the dispersion of these branches for different applied voltages of $-210$, 110 and 210 V respectively, as measured by taking successive transmission spectra for varying cavity lengths. As the cavity length is decreased from $L = 0.56$ μm to $L = 0.41$ μm the cavity mode energy increases according to $E_c = \frac{qhc}{2L_o}$, where $q = 4$ is the longitudinal mode index and $L_o$ is the optical cavity length (white continuous line, $\times$). As it traverses the exciton energies at $E_{WM}$ (white continuous line, $\ast$) and $E_F$ (white continuous line, $\Delta$), an anti-crossing is visible which is indicative of the strongly coupled nature of the system. In this way, coupled eigenstates of equation (2) are formed which we call the lower, middle and upper polariton branch (LP, MP and UP). The constituent uncoupled components for these branches quantified by $\alpha^2$, $\beta^2$ and $\gamma^2$ are shown in Fig. 2d–f. The nature of the LP (UP) branch transits from photonic (F-excitonic) to WM-excitonic (photonic) as the cavity length is decreased. The more interesting MP branch changes from WM-excitonic to F-excitonic nature for decreasing mirror separation. For cavity lengths in the region 0.525 μm $> L >$ 0.425 μm, its photonic fraction increases and

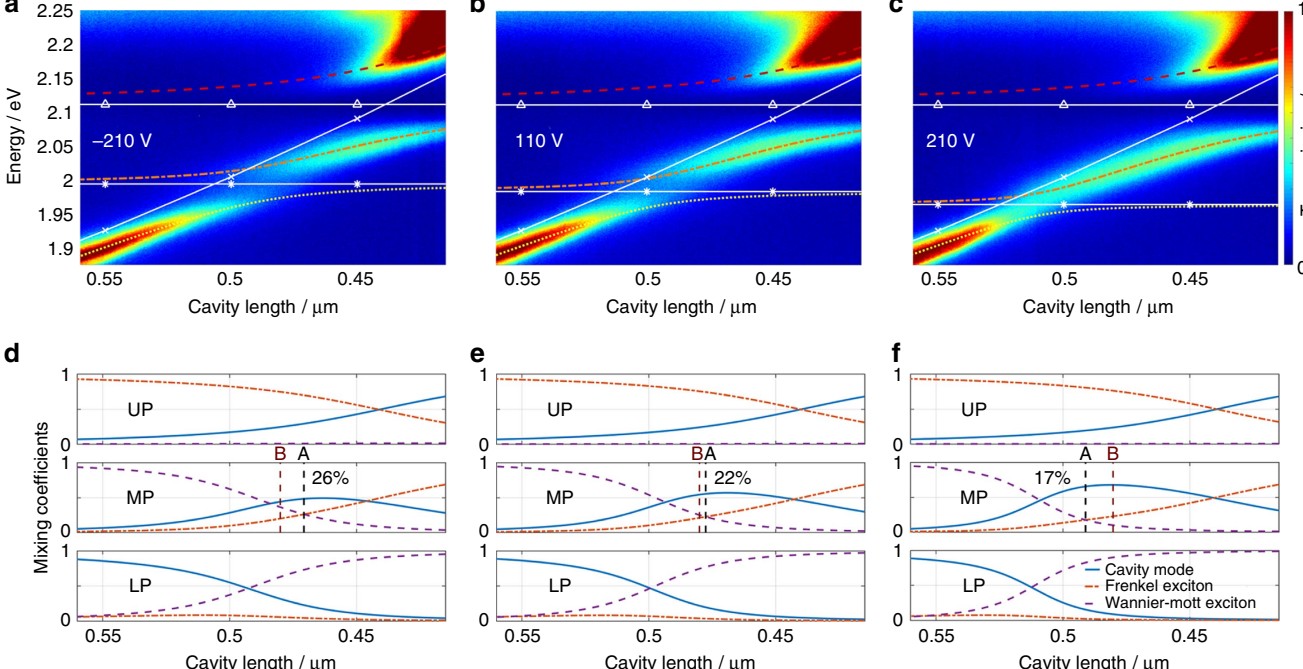

**Figure 2 | Electrically controlled hybridization of Frenkel- and Wannier-Mott-excitons in a polariton state. (a–c)** Successive transmission spectra of hybrid WS$_2$-TDBC microcavity for decreasing cavity length from left to right and different applied voltages of $-210$, 110 and 210 V for **a–c**, respectively. The white, continuous lines correspond to the uncoupled energies of Frenkel-exciton (TDBC, triangles), Wannier-Mott exciton (WS$_2$, stars) and cavity mode (crosses). The dashed lines in colour show the dispersion for the coupled system consisting of the three polariton branches, lower polariton (LP, yellow, – -), middle polariton (MP, orange, — –) and upper polariton (UP, red, — —), where the terms in brackets denote the name of the state, the line colour and the line style respectively. **(d–f)** Photonic (cavity mode, blue, continuous), Frenkel-excitonic (TDBC, red, — -) and Wannier-Mott-excitonic (WS$_2$, purple, — —) contribution to the three polariton branches LP, MP and UP for the dispersions plotted above respectively. Two points A (black dashed line) and B (red dashed line) mark cavity lengths at which: (A) Frenkel- and Wannier-Mott-exciton contribution to the middle polariton branch is equal, (B) the cavity length is $L = 0.48$ μm and the dominant exciton contribution can be swapped electrically. The numerical value displayed in the MP panel as a percentage gives the value of Frenkel- and Wannier-Mott-exciton contribution $\beta^2$ and $\gamma^2$ at point A.

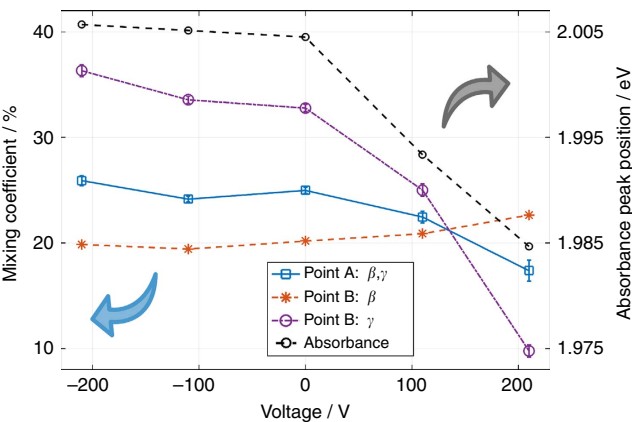

**Figure 3 | Electrical control over polariton composition.** Frenkel- and Wannier-Mott-exciton fraction $\beta$ and $\gamma$ in hybrid polariton state for different applied voltages (colour, symbols, left ordinate, blue arrow) and interpolated absorbance peak position of $WS_2$ outside the cavity (black, dashed, right ordinate, grey arrow). The exciton fractions are shown for two points, where point A corresponds to the cavity length at which $\beta = \gamma$ (maximal mixing) and point B to a fixed cavity length of $L = 0.48\,\mu m$. The errorbars were obtained by fitting the polariton dispersion (equation (2)) to the transmission data and are of similar size as the symbols. The absorbance peak positions were acquired by fitting a Gaussian lineshape to the absorbance shown in Fig. 1. More information about the error estimation is given in Supplementary Fig. 6 and Supplementary Note 4.

together with the two excitons, a hybrid polariton state is formed. We shall define two points of interest along the dispersion curve: Point A corresponds to the cavity length at which the middle polariton branch has equal weights of F- and WM-exciton, and is labelled in Fig. 2d–f (middle panels). Point B stands for a fixed cavity length of $L = 0.480\,\mu m$, which we will show corresponds to a cavity length where we can electrically switch the MP state from having a dominant WM exciton component to a dominant Frenkel exciton component.

**Electrical control over polariton composition.** By applying an external electric field, $E_{WM}$ and the corresponding absorption amplitude and linewidth are modified leading to a change in the polariton mixing coefficients. As the applied voltage is increased from $-210\,V$ to $110$ ($210\,V$), the Rabi splitting about the WM exciton decreases from $2V_{WM} = 57$ to $46\,meV$ ($35\,meV$) and $E_{WM}$ shifts from $1.997$ to $1.987\,eV$ ($1.968$) $eV$. The absorption linewidth of the WM-excitonic transition increases such that the splitting is not resolvable for applied voltages above $150\,V$ (Fig. 2c). The Rabi splitting about the Frenkel exciton energy is $2V_F = 114\,meV$ for all applied voltages. For the same change in voltage the excitonic weight at point A changes from $\beta^2 = \gamma^2 = 26\%$ to $22\%$ ($17\%$) while the photonic fraction increases from $\alpha^2 = 48\%$ to $56\%$ ($66\%$). At the same time the cavity length corresponding to point A shifts from $L = 0.471\,\mu m$ to $0.478\,\mu m$ ($0.492\,\mu m$). At point B the composition of the polariton swaps from $\beta^2 = 20\%$, $\gamma^2 = 36\%$ to $\beta^2 = 21\%$, $\gamma^2 = 25\%$ ($\beta^2 = 23\%$, $\gamma^2 = 10\%$) while $\alpha^2$ increases from $44$ to $54\%$ ($67\%$) as the applied voltage is changed from $-210$ to $110\,V$ ($210\,V$), therefore the dominant exciton component within the MP branch switches from WM to Frenkel with increasing voltage. Figure 3 shows the changes in polariton composition as a function of the applied voltage for more intermediate voltages. The black dashed line is obtained by fitting the absorption peaks presented in Fig. 1d with a Gaussian lineshape and interpolating the central energy (right ordinate). The datapoints quantifying the polariton composition are the result of fits of the polariton dispersion given by

equation (2) to the transmission data shown in Fig. 2a–c (Supplementary Fig. 6). More details on the fitting procedure and the origin of the errorbars is given in Supplementary Note 4. It is evident that the change in the hybrid polariton composition stems from the altered WM-exciton state, which can be controlled electrically in the above stated manner. We present another dataset acquired from a different $WS_2$ flake in the Supplementary Note 5 and Supplementary Figs 7 and 8.

## Discussion

The change in absorption peak position and lineshape in response to the change in applied voltage originates from the local change in electron density. This effect has been described previously for the PL lineshape[11,14,22] and more recently for the absorption spectrum of TMDs[23,24]. It is attributed to a combined result of Coulomb scattering, Pauli blocking and Coulomb screening which causes a transfer of oscillator strength from neutral ($X^0$) to charged ($X^-$) exciton together with a shift in energy of both states (see refs 23,24 for a more detailed description). In our system with a laterally applied field without direct carrier injection we make use of the abundance of electrons commonly found in $WS_2$ and $MoSe_2$ flakes[14,22], whose distribution across the flake can be altered with the electric field (Supplementary Fig. 3 and Supplementary Note 3). Due to the geometry of our system, the steady state of this distribution is reached on a millisecond timescale. We attribute this slow speed to the occurrence of scattering defects in the transferred monolayer, caused by locally induced strain and impurities reducing the mobility. Improvements in sample preparation such as the embedding of single TMDC layers within quasi non-interacting hBN heterostructures together with a back-gated electric field geometry would allow for fast switching times, limited with the current technology by the high resistance and therefore the high RC constant of the measurement circuit[16]. Advances in contacting two-dimensional semiconductor layers up to the ohmic contact would remedy this situation[15].

While the transition from $X^0$ to $X^-$ with increased electron density is well understood[21,23], the impact such change has on exciton-polariton states is non-trivial and the topic of current research[24,25]. Due to the fermionic nature of a trion state, the trion-trion interaction is stronger than the purely exciton mediated non-linearity[26], which would make trion-polariton systems attractive for observing strongly correlated phenomena obeying Fermi-Dirac statistics.

In addition to the electrical tuning of the polariton state, the open cavity allows to bring higher longitudinal modes with decreasing exciton–photon coupling into resonance[20], which results in different mixing of the polariton components.

In summary, we have fabricated an open microcavity system containing a monolayer of transition metal dichalcogenide and layer of J-aggregate dye that are simultaneously strongly coupled to a cavity mode at room temperature. The resulting hybrid polariton states have a mixed F/WM exciton nature. Application of a transverse electric field across the monolayer results in a shift of the absorption peak energy and allows controllable tuning of the exciton mixing within the hybrid polariton states. Polariton-polariton interactions give rise to non-linear effects, rendering polaritonic systems attractive to observe a multitude of fascinating phenomena such as inversionless lasing, superfluidity and topologically non-trivial states[27]. These interactions are weaker for localized Frenkel excitons than for WM excitons typical in a crystalline lattice. Our results show how the hybridization between such distinct excitons could be controlled electrically at room temperature. These findings could open pathways to novel photonic devices with engineered optical properties.

## Methods

**Sample preparation.** The open microcavity consists of two opposing flat mirrors, a large dielectric DBR with 10 pairs of $SiO_2$, $TiO_2$ with central wavelength of $\lambda = 640$ nm and a smaller silver mirror (Fig. 1b). To enable electrical control within the cavity, silver electrodes with a width and a spacing of 90 µm are thermally evaporated on top of the DBR (Supplementary Fig. 1). The thickness of these electrodes is similar to the thickness of the silver layer on the small opposing mirror, approximately 50 nm. The $WS_2$ flakes are grown as described in ref. 19 and transferred onto the dielectric mirror stack, which has a low refractive-index terminated configuration to provide an anti-node of the electric field at the mirror surface and thus optimal coupling to the monolayer (Supplementary Fig. 2). The distribution of $WS_2$ flakes relative to the electrodes is random, with 90% of the flakes overlapping partially with the silver electrodes. The results presented here were obtained from a flake which was not in physical contact with the electrodes, thus ensuring purely electrostatic tuning (Supplementary Fig. 3). The other component of the hybrid system is the organic J-aggregated dye TDBC. After dissolving in a gelatine-water solution, the dye was spin-coated onto the small silver mirror to give a polymer-dye layer of about 300 nm thickness. The absorbance of the TDBC film can be tuned by varying the concentration of the dye (Supplementary Figs 4 and 5). For a more detailed exposition of the sample preparation we refer the reader to the Supplementary Note 1a.

**Measurements.** The small silver mirror with the dye was mounted on a three-dimensional piezo actuated stage, which makes positioning of the silver mirror relative to the $WS_2$ flake possible and allows electrical control of the cavity length. By moving the silver plinth over a region of the DBR mirror which holds monolayer $WS_2$ and reducing the distance between the two mirrors below $L \approx 5$ µm stable cavity modes interacting strongly with the $WS_2$ excitons appear. The mode structure of the system can be observed by measuring the transmitted light of a spectrally broad lightsource. For this analysis the light was focused onto an Andor combined spectrometer/CCD. For more details on the measurements we refer the reader to the Supplementary Note 1b. In addition, we discuss the optical properties of the individual components of the hybrid polariton system in Supplementary Note 2.

**Data availability.** The data that support the findings of this study are available at the Oxford University Research Archive (https://ora.ox.ac.uk/).

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

## Acknowledgements

We thank Radka Chakalova at the Begbroke Science Park for helping with the thermal evaporation and dicing of the mirrors. L.C.F. acknowledges funding from the Leverhulme Trust. J.M.S. and D.M.C. acknowledge funding from the Oxford Martin School and EPSRC grant EP/K032518/1. R.A.T. also acknowledges funding from the Oxford Martin School. D.G.L. and D.M.C. acknowledge EPSRC grant EP/M025330/1.

## Author contributions

Z.H. and J.H.W. grew the $WS_2$ and deposited it on the mirror. D.M.C. prepared the organic dye, measured the Rabi splitting as a function of the concentration and coated the small mirror with it. All other sample preparation steps and measurements were carried out by L.C.F. under the supervision of R.A.T. and J.M.S. All authors contributed to the preparation of the manuscript.

## Additional information

**Competing financial interests:** The authors declare no competing financial interests.

