## [Peer Review File · Nature Communications]

Reviewers' Comments:

Reviewer #1 (Remarks to the Author)

The authors present an experimental study on cavity exciton-polariton effects of a hybrid structure containing a WS₂ monolayer and organic dye. The electric field tuned coupling of Wannier-Mott-type excitons in the WS₂ monolayer and photons, as well as the coupling of Frenkel excitons in the organic layer and photons, were clearly demonstrated. These results may be useful for the electrical control of different exciton polaritons, however, in terms of physics of cavity polariton, there is nothing new here. Both Wannier-Mott-type and Frenkel type polaritons have been intensively studied and well understood in various two-dimensional transition metal dichalcogenides and organic materials in recent years (See papers in Nature Photonics, Nature Materials etc). Besides, the polaritons are optically injected here. For opto-electronic device application, electrical injection is preferable. Therefore, it doesn't meet the criteria of Nature Comm..

The followings are several minor technical questions:

1. In order to electrically tune the hybrid polaritons, a 90 nm periodical silver electrodes were deposited on the top of DBR structure, how will these electrodes affect the light field distribution in the open cavity?

How does the external field affect the Frenkel excitons in the organic layer? Do the authors observe both X₀ and X₋?

Reviewer #2 (Remarks to the Author)

Organic-inorganic hybrid materials are an exciting playground on the search for new phenomena at the respective interfaces. This is in particular so what concerns optical excitations. Excitons are typically strongly bound in organic materials (Frenkel type) while the electron-hole interaction is well screened and thus weak in inorganic systems (Wannier type). Thus one can expect different types of excitations arising at hybrid interfaces. Hybrid excitons can have different character, with either hole (electron) residing in the organic (inorganic) component or being composed from hybridized states. The former are a desired feature in hybrid solar cells, facilitating electron-hole separation at the interface.

The current work pursues an alternative route by coupling excitons from either side through an exciton polariton state that can be tuned by an applied voltage. This is a fascinating idea, that is not only very interesting from a fundamental scientific point of view but also promising with respect to applications in opto-electronics. I am not an expert in the experimental setup and thus can't judge about this. Apart from this being assessed by an expert, I support publication of this manuscript.

Reviewer #3 (Remarks to the Author)

The paper "Electrically tunable organic-inorganic hybrid polaritons with monolayer WS₂" by Flatte et al. presents new interesting results for a fabricated device that consists of an open microcavity containing both organic dye and a monolayer of the transition metal dichalcogenide WS₂. These materials are coupled strongly to a cavity mode, and hybridized polariton states that are formed, combining properties from both materials. The resulting hybrid polariton states have a mixture of the different types of excitons that have Frenkel and Wannier-Mott exciton nature. Application of a transverse electric field across the monolayer results in a shift of the absorption peak energy and allows controllable tuning of the exciton mixing within the hybrid polariton states. The article is well written and references reflect the current status of the research in the field.

I recommend this manuscript for publication when my concern will be addressed. My minor concern related to the abstract of the manuscript. As the abstract is written, it is strongly related to the cited references. There are 28 references in manuscript and 16 of them are cited in the abstract. I'll suggest to make abstract shorter with minimum citations.

Reviewer 1:

The authors present an experimental study on cavity exciton-polariton effects of a hybrid structure containing a WS2 monolayer and organic dye. The electric field tuned coupling of Wannier-Mott-type excitons in the WS2 monolayer and photons, as well as the coupling of Frenkel excitons in the organic layer and photons, were clearly demonstrated. These results maybe useful for the electrical control of different exciton polaritons, however, in terms of physics of cavity polariton, there is nothing new here. Both Wannier-Mott-type and Frenkel type polaritons have been intensively studied and well understood in various two-dimensional transition metal dichalcogenides and organic materials in recent years (See papers in Nature Photonics, Nature Materials etc). Besides, the polaritons are optically injected here. For opto-electronic device application, electrical injection is preferable. Therefore, it doesn't meet the criteria of Nature Comm..

Thank you for your remark. Over the last decade there have been indeed many reports of interesting polariton phenomena in high impact journals, such as polariton condensation, electrically injected polariton lasing and polaritonic solitons. Most of these reports have been obtained at cryogenic temperatures in monolithically grown samples, only controllable by the optical excitation intensity. Our work adds to this rich base of knowledge by a) demonstrating the direct electrical control over a (hybrid) polariton state independent of the optical excitation and b) showing room-temperature operation of a highly versatile, easy to integrate and scalable (heterostructures of two-dimensional materials) polariton device. We note as well, that many important optoelectronic devices are not electrically driven (e.g. switches for optical communications).

In order to electrically tuned the hybrid polaritons, a 90 μm periodical silver electrodes were deposit on the top of DBR structure, how will these electrodes affect the light field distribution in the open cavity?

As you correctly remarked the spacing between the electrodes is 90 μm . In fact a cavity mode in the region of these electrodes would be spectrally broader and the number of round-trips would be reduced due to the additional absorption of the second silver interface. For the data that we present in the manuscript this effect and its relevance for our conclusions can be estimated:

1) Given the cavity length of around 500 nm and the reflectivity of the silver mirror $R = 0.95$, we can equate an effective mode area $A = \frac{\pi L \lambda}{1-R} \approx 19.5 \mu\text{m}^2$, as derived in ([R1] K. Ujihara, Japanese Journal of Applied Physics, vol. 30, pp. L901-L903, 1991). This area translates to a mode radius of $r \approx 2.5 \mu\text{m}$. Since the region of the sample from which we obtained the results is more than 15 μm from the next silver electrode, the effect from the electrode on the cavity mode is negligible.

2) The way we obtained our data ensured that any global effect on the cavity modes would not hinder our conclusions: The two parameters that we varied were the cavity length and the applied electric field. The change in cavity length might lead to a change in mode size (as outlined in 1)), which could affect the quality factor of the mode. This effect would thus be visible in the linewidth of cavity modes, which

would increase for decreasing cavity lengths, which is not the case. The applied electric field depends on the geometry of the electrodes but remains unchanged for different field strengths as the position from where we obtained the data on the sample stayed fixed.

We have incorporated parts of this answer in the supplemental materials App. A.

How does the external field affect the Frenkel excitons in the organic layer?

The j-aggregate exciton energy can be shifted in an electrical field through the Stark effect, but it requires field strengths of about 10^6 V/cm for a 20 nm shift (as determined through electroabsorption measurements - unpublished), whereas the field strengths that we obtained in our experiment were about 2.33×10^4 V/cm. Our data gives experimental evidence of this, since the Frenkel exciton energy remained unchanged (within the uncertainty of the Lorentzian lineshape fit) for any applied electric field. We also note that, assuming a dielectric constant of 10 and a binding energy of 1 eV (typical of organic materials), an electric field of $\approx 1.4 \times 10^6$ V/cm would be required to disassociate the excitons (see e.g. Handbook of Conducting Polymers, ed. Skotheim, Elsenbaumer & Reynolds, 2nd ed. 1998, pub. Marcel Dekker). We have incorporated most of this answer in the the Suppl. Mat. App. A.

Do the authors observe both X0 and X-?

The occurrence of the charged exciton, or trion (X^-), peak depends on the density of free electrons in the material (as explained in e.g. [R2] Plechinger et al., Phys Status Solidi RRL 9, 457-461 (2015) or [R3] Zhu et al., Scientific Reports 5, 9218 (2015)). As Fig. 2(a) in Ref. [R3] shows, the trion can be quenched by applying a negative gate voltage (this is exactly the effect that we use for the tunable hybridisation in the paper). The zero bias level of weight in neutral and charged exciton is dependent on the intrinsic doping of the WS_2 flake, which is thought to be a result of the growth and transfer method. On our CVD grown samples we see variations of X^- contribution on one flake and even larger variations when comparing multiple flakes. In general the X/X^- ratio found for our sample is similar to bias values between -30 V and 0 V in Fig. 2(a) in Ref. [R3]. To demonstrate the strong coupling to a cavity mode we chose a region with minimal trion contribution (as visible in the absorption lineshape in Fig. 1 d) for negative applied voltages). The reasoning behind this was, that with more trionic contribution the polariton linewidth increases, the asymmetry between upper and lower polariton branch increases and the splitting is less obvious. We added most of this answer to our description of the electrical tuning in the the Suppl. Mat. App. C. Thank you for the interesting questions.

Reviewer 3:

My minor concern related to the abstract of the manuscript. As the abstract is written, it is strongly related to the cited references. There are 28 references in manuscript and 16 of them are cited in the abstract. I'll suggest to make abstract shorter with minimum citations.

Thank you for the suggestion. Indeed a reduction in the number of references increases the focus of our exposition. In light of this, we have removed 5 references from the abstract and reduced the total number of references for the whole paper to 24.

Reviewer #1 (Remarks to the Author)

The authors have answered all my questions and addressed my concerns nicely. I am now happy to recommend the publication of their paper in Nature Communications.